# SRDIFFUSION: ACCELERATE VIDEO DIFFUSION INFERENCE VIA SKETCHING-RENDERING COOPERATION

## ABSTRACT

Leveraging the diffusion transformer (DiT) architecture, models like Sora, CogVideoX and Wan have achieved remarkable progress in text-to-video, image-to-video, and video editing tasks. Despite these advances, diffusion-based video generation remains computationally intensive, especially for high-resolution, long-duration videos. Prior work accelerates its inference by skipping computation, usually at the cost of severe quality degradation. In this paper, we propose SRDiffusion, a novel framework that leverages collaboration between large and small models to reduce inference cost. The large model handles high-noise steps to ensure semantic and motion fidelity (Sketching), while the smaller model refines visual details in low-noise steps (Rendering). Experimental results demonstrate that our method outperforms existing approaches, over $3\times$ speedup for Wan with nearly no quality loss for VBench, and $2\times$ speedup for CogVideoX. Our method is introduced as a new direction orthogonal to existing acceleration strategies, offering a practical solution for scalable video generation.

## 1 INTRODUCTION

With the rapid development of diffusion models, they have become the mainstream approach for generating high-quality images, audio, and video. Among them, DiT-based (Peebles & Xie, 2023) video generation models have also advanced rapidly, including Sora (Brooks et al., 2024), CogVideoX (Yang et al., 2024), OpenSora (Zheng et al., 2024), Wan (Wang et al., 2025), and others. These models have been widely applied in various tasks such as image-to-video generation, text-to-video generation, video editing (Wang et al., 2023a; Jiang et al., 2025), and video personalization (Wei et al., 2024).

However, despite significant advancements in generation quality, diffusion-based video generation remains computationally expensive and time-consuming. The inference cost increases rapidly with model size, video resolution, and temporal duration. For instance, generating a 5-second 720p video using the Wan-14B model can take nearly an hour on a single NVIDIA A100 GPU. Prior acceleration works (Lv et al., 2024; Zhao et al., 2024b; Liu et al., 2024) focus on the *computation skipping* methods, caching certain diffusion steps or intermediate results to exploit similarities across different sampling stages. Despite their limited speedup, they often lead to a noticeable decline in generation quality.

In this study, based on observations from the VBench evaluation of both large and small models, we find that the primary advantage of large models lies in their superior semantic capabilities, particularly in following instructions for composition and motion. However, the difference is much smaller in terms of detail quality (the "quality" dimension in VBench). On the other hand, small models have a significant advantage in runtime efficiency.

Building on these insights, we propose SRDiffusion, a novel approach to accelerate diffusion inference via sketching-rendering cooperation. Specifically, SRDiffusion will use large model during the high-noise steps to generate higher-quality structure and motion that better align with textual instructions (*Sketching*), while use the small model during the low-noise steps to generate finer details (*Rendering*), thereby accelerating the overall diffusion process. In addition, we design a metric to dynamically determine the switch from the sketching phase to the rendering phase, enabling a more flexible and guaranteed speed-quality trade-off.

The contributions of our paper are as follows:

- We reveal the distinct trade-offs in semantics, quality, and speed between large and small models, and highlight the potential for their cooperative use.
- The introduction of sketching-rendering cooperation, a novel approach that leverages large models for sketching and small models for rendering, to accelerate video diffusion.
- Sketching-rendering cooperation operates as a plug-and-play solution and can be seamlessly integrated with various optimization techniques, such as caching mechanisms, system-level enhancements and distillation model.
- We design an adaptive switching metric to decide the time of switch from the sketching phase to rendering phase.
- Experimental results demonstrate that our method outperforms existing approaches, over $3\times$ speedup for Wan with nearly no quality loss for VBench, and $2\times$ speedup for CogVideoX.

## 2 PRELIMINARIES AND RELATED WORKS

### 2.1 DIFFUSION PROCESS.

Diffusion models (Ho et al., 2020) simulate the gradual diffusion of data into Gaussian noise (the forward process) and the subsequent recovery of the original data from noise (the reverse process) to achieve the modeling and generation of complex data distributions. In the forward diffusion process, starting from a data point sampled from the real distribution, $x_0 \sim q(x)$, Gaussian noise $\epsilon_t$ is gradually added over $T$ steps to produce a sequence of increasingly noisy samples $\{x_t\}_{t=1}^T$, where the noise level is controlled by $\alpha_t$:

$$x_t = \sqrt{\alpha_t}x_{t-1} + \sqrt{1-\alpha_t}\epsilon_t, \quad \epsilon_t \sim \mathcal{N}(0, I), \quad \alpha_t \in [0, 1], \quad t = 1, 2, ..., T \quad (1)$$

In the reverse diffusion process, a neural network is trained to approximate the conditional distribution $O(x_t, t)$, effectively learning how to denoise a sample at each step. The model iteratively removes noise, moving from $x_T$ back to a clean sample $x_0$, thereby generating new data consistent with the training distribution. A scheduler $\Phi$ determines how to exactly compute $x_{t-1}$ from $x_t$, $t$ and the output of the neural network $O(x_t, t)$: $x_{t-1} = \Phi(x_t, t, O(x_t, t)), \quad t = T, ..., 2, 1.$

### 2.2 VIDEO DIFFUSION TRANSFORMER

The video diffusion transformer consists of three primary components: a 3D Variational Autoencoder (3D VAE), a text encoder, and a diffusion transformer. The 3D VAE compresses the input video from the pixel space $V \in \mathbb{R}^{(1+T)\times H \times W \times 3}$ into a latent representation $x \in \mathbb{R}^{(1+T/C_T)\times H/C_H \times W/C_W}$, where $C_T, C_H, C_W$ denote the compression rates in the temporal, height, and width dimensions, respectively. This latent representation is then reshaped into a flattened sequence $z_{vision}$. The text encoder processes the input text into a corresponding latent embedding $z_{text}$. To embed text conditions, Wan uses cross-attention, whereas CogVideoX concatenates $z_{vision}$ and $z_{text}$ directly.

### 2.3 DIFFUSION INFERENCE ACCELERATION

To accelerate diffusion inference, several studies have focused on designing more efficient schedulers, such as DDIM (Song et al., 2020). Others have explored advanced ODE or SDE solvers to improve sampling efficiency (Karras et al., 2022; Lu et al., 2022b;a). In parallel, model distillation approaches aim to reduce inference time by training smaller models or models that require fewer sampling steps. For instance, (Wang et al., 2023b; Salimans & Ho, 2022) employ distillation techniques to achieve high-quality generation with only a few steps. Additionally, some research efforts focus on improving model architecture (Xie et al., 2024; Chen et al., 2024) or generative paradigm (Tian et al., 2024; Gu et al., 2024; Zhang & Agrawala, 2025; He et al., 2025) to enhance efficiency. However, these methods typically require fine-tuning or additional training, incurring extra computational costs.

Training-free methods can be broadly categorized into two strategies: skip-computation and system-level optimizations. Skip-computation techniques exploit redundancy across sampling steps, often

leveraging caching mechanisms to accelerate inference (Lv et al., 2024; Zhao et al., 2024b; Liu et al., 2024). For example, T-GATE (Zhang et al., 2024b) caches self- and cross-attention outputs at various stages, while PAB (Zhao et al., 2024b) selectively caches and broadcasts intermediate features based on attention block characteristics. TeaCache (Liu et al., 2024) introduces timestep embedding-aware caching to bypass certain diffusion steps. On the other hand, system-level approaches, such as quantization (Zhang et al., 2024a; Zhao et al., 2025; Li et al., 2024b), sparsity attention (Team, 2024; Zhang et al., 2025b) and parallelism (Fang & Zhao, 2024; Fang et al., 2024; Li et al., 2024a; Zhao et al., 2024a), aim to further reduce computational overhead through architectural and execution-level optimizations.

In addition to training-free acceleration methods, many studies focus on distilling a faster model. For example, several works use distribution matching distillation to reduce the number of steps in the final model's denoising process, including CausVid (Yin et al., 2025) and FastWan (Zhang et al., 2025a; Team, 2024). Notably, FastWan further introduce sparse attention during distillation.

This work proposes a new optimization direction that is orthogonal to the related studies mentioned above: leveraging collaboration between large and small models for diffusion inference. The cooperation concept is widely discussed in the context of LLM serving, known as speculative decoding (Leviathan et al., 2023; Chen et al., 2023). In contrast to auto-regressive models, where a smaller model is used for speculation and a larger one for verification, we propose the opposite for diffusion models: employing a larger model for sketching and a smaller model for rendering. The optimization principles are also quite different: speculative decoding improves the hardware utilization of large models through batch verification, whereas our method directly reduces computation by using a smaller model for certain steps. And our proposed approach aligns with certain edge-cloud system architectures, such as Hybrid-SLM-LLM (Hao et al., 2024) and HybridSD (Yan et al., 2024), where a lightweight model on the edge collaborates with cloud-based models to reduce the overall generation cost.

## 3 METHOD

### 3.1 MOTIVATION

Based on evaluation results from VBench (Huang et al., 2024) across both the semantic and quality dimensions, we observe that the Wan 14B model demonstrates a significant improvement in the semantic dimension compared to the Wan 1.3B model. In terms of quality, however, the two models achieve relatively close scores. In our practical usage, we also find that Wan 14B follows instructions more effectively and exhibits clearly superior compositional capabilities. Nevertheless, there is a notable trade-off in inference latency: the Wan 1.3B model requires only 158,s per sample, which is over five times faster than the Wan 14B model at 841,s. [1]

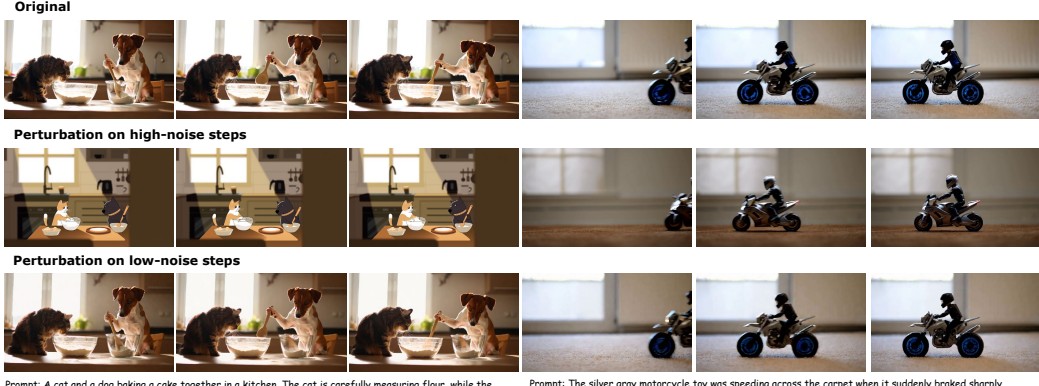

Figure 1: Impact of perturbations at various diffusion steps on the quality of video frames.

---

[1]Latency is measured on a public cloud A800 instance.

To identify the most critical phase of the diffusion process for capturing semantics, we introduce perturbations in the form of biased Gaussian noise into the latent representation at different stages. As shown in Figure 1, perturbations introduced during the early high-noise steps (steps 0 to 10) lead to significant semantic changes, altering the overall structure and style of the video. In contrast, perturbations applied during the later low-noise steps (steps 10 to 50) result in only subtle variations in fine-grained details, such as the background chair in the first example or texture refinement in the second, while largely preserving the core semantics.

## 3.2 SKETCHING-RENDERING COOPERATION

Our previous analysis highlighted that the early high-noise steps of the diffusion process are particularly important for semantic aspects such as composition and motion. During this phase, larger models demonstrate significantly stronger semantic capabilities compared to smaller ones. In contrast, the low-noise steps mainly focus on fine-grained details. Although smaller models are slightly less capable in this stage, they offer a clear advantage in terms of speed. Based on these observations, we propose the Sketching-Rendering Cooperation framework, which is illustrated in the overall architecture of Figure 2.

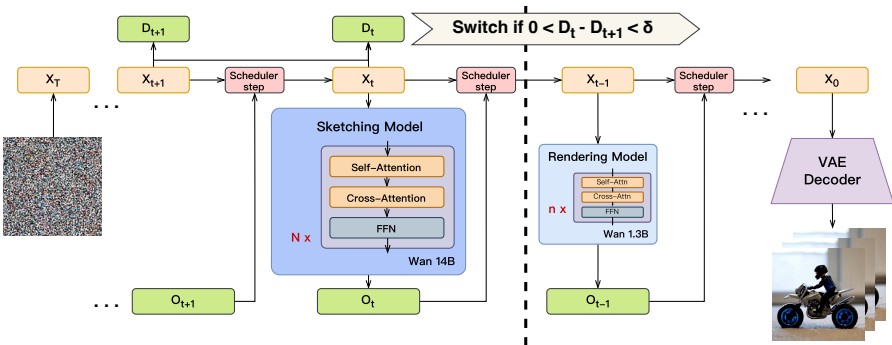

Figure 2: Overview of Sketching-Rendering Cooperation. Taking pipeline of Wan model as an example, illustrates the pipeline switches from Wan 14B to Wan 1.3B at timestep $t$.

In this pipeline, the entire diffusion process generates a video based on the given input prompt. The prompt is first processed by a text encoder, and its encoded representation is used as a condition throughout every step of the diffusion process. The generation begins with a randomly initialized noise latent, which is initially handled by the sketching model. This model predicts the noise and updates the latent using a scheduler step. Then, an adaptive switching mechanism will determine whether to continue using the sketching model or switch to the rendering model. At a certain timestep $t$, once the mechanism decides that the rendering model can take over, the remaining diffusion steps are performed by it. At the end, the resulting latent is decoded into a video using a 3D VAE decoder.

Throughout the process, the collaboration between the sketching model and the rendering model ensures a balance between quality and efficiency. The sketching model preserves high-level semantics in the early phase, while the rendering model generates detailed content in the later phase with lower computational cost.

In most state-of-the-art models, such as Wan (Wang et al., 2025), Hunyuan (Kong et al., 2024), the 3D VAE is typically trained separately before training the DiT. Within the same model family, different model sizes generally share the same VAE, which uses identical compression parameters and maintains a consistent latent space. In this case, when switching from the sketching model to the rendering model, the latent tensor shapes remain the same, and no additional alignment is required. The corresponding pseudocode is provided in Algorithm 2. The pseudocode shows the sketching-rendering cooperation for Wan with classifier-free guidance.

---

**Algorithm 1** Diffusion Inference Process for Sketching-Rendering Cooperation for Wan.

---

1: Initialize latent variable $\mathbf{z}$
2: Set initial model: ExecModel ← SketchingModel
3: **for** each timestep $t$ in $\{T, \ldots, 1\}$ **do**
4:     Predict conditional noise: $\hat{\epsilon}_{\text{cond}}$ ← ExecModel($\mathbf{z}, t$, condition)
5:     Predict unconditional noise: $\hat{\epsilon}_{\text{uncond}}$ ← ExecModel($\mathbf{z}, t$, no condition)
6:     Apply guidance: $\hat{\epsilon}$ ← $\hat{\epsilon}_{\text{uncond}} + s \cdot (\hat{\epsilon}_{\text{cond}} - \hat{\epsilon}_{\text{uncond}})$
7:     Update latents: $\mathbf{z}$ ← SchedulerStep($\hat{\epsilon}, t, \mathbf{z}$)
8:     **if** ExecModel is SketchingModel and switch_condition($\mathbf{z}$) **then**     ▷ Switch Condition see Sec. 3.3
9:         Switch to rendering model: ExecModel ← RenderingModel
10:    **end if**
11: **end for**
12: **return** Final generated sample $\mathbf{z}_0$

---

Besides cooperation within same model family which share the same VAE, sketching-rendering cooperation framework also can work for models with different VAE, which we called **cross-family cooperation**. In this case, we cannot directly transfer the latent representation from the sketching model to the rendering model due to their misaligned latent spaces. Instead, we use the predicted clean sample $x_0$ at step $i$, decode it into pixel space with the sketching model's VAE, and then re-encode it with the rendering model's VAE to obtain a latent representation in the rendering model's space. After that, we add the corresponding noise level. The signal-to-noise ratio is used to determine the step at which the rendering model should take over. The detailed algorithm and pseudocode are provided in the Appendix A.2.

### 3.3 ADAPTIVE SWITCH

To ensure a consistent effect for different prompts, we dynamically determine the optimal switching step from the sketching model to the rendering model. Following Liu et al. (2024) and Wimbauer et al. (2024), we observe that the predicted Gaussian noise change diminishes during the diffusion process, and the second-order derivative of this change continuously decreases and gradually stabilizes. We use the relative L1 distance to characterize the difference of the denoised sample across steps as follows:

$$D_t = tanh(\frac{||x_t - x_{t+1}||_1}{||x_{t+1}||_1}) \tag{2}$$

where $x_t$ indicates the denoised sample at timestep $t$ following 1 and $tanh$ is applied to scale the absolute value to the range of (0,1). The denoised sample changes $D_t$ of multiple prompts throughout the diffusion process in Wanx 14B and CogVideoX 5B are illustrated in Figure 3(a)(c), and the their second-order derivative are illustrated in Figure 3(b)(d), respectively.

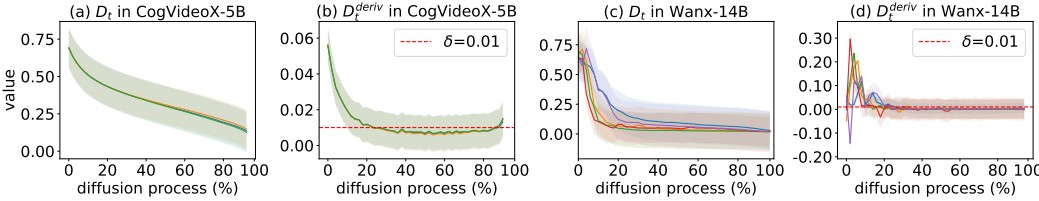

Figure 3: Predicted noise difference across denoising steps in Wan-14B 480p and CogVideoX-5B 480p. Different colors represent the value of different prompts.

During runtime, we record the second-order derivative of predicted noise $D_t^{deriv}$ of each timestep and compare it with the threshold $\delta$. Besides, to ensure the video quality, we set `FIX_STEP` as the minimum scheduler steps to execute the sketching model, which is set to 5 in our experiments. Note that as the reverse diffusion process progresses, the index of denoising timestep $t$ decreases, while that of the scheduler step $\tau$ increases. We switch to the rendering model once second-order derivative satisfies $0 < D_t^{deriv} = D_t - D_{t+1} < \delta$ and the timestep $\tau$ exceeds `FIX_STEP`.

In practice, we provide the distribution of steps selected by the adaptive switching strategy under different values of $\delta$ in the Appendix A.5.

## 4 EXPERIMENTS

### 4.1 EXPERIMENTAL SETTING

**Models.** We conduct experiments on multiple video generation models, including Wan and CogVideoX, which provide two model sizes: Wan includes 14B and 1.3B variants, while CogVideoX includes 5B and 2B variants. As model families, they share the same VAE within each family.

**Baselines.** For baseline methods, we select PAB (Zhao et al., 2024b) and TeaCache (Liu et al., 2024), both of which are specifically designed to accelerate video diffusion models through caching mechanisms. These techniques are conceptually similar to our approach in that they aim to skip redundant computations, whereas our method switches to a smaller model for computation reduction.

**Metrics.** For quality evaluation, we utilize VBench (Huang et al., 2024) to evaluate the generation quality. We use VBench standard prompt set and generate 5 videos with different seeds for each prompt. In addition, we report standard perceptual and pixel-level similarity metrics, including Learned Perceptual Image Patch Similarity (LPIPS) (Zhang et al., 2018), Structural Similarity Index Measure (SSIM) (Wang et al., 2004), and Peak Signal-to-Noise Ratio (PSNR). For efficiency evaluation, we measure the inference latency per sample as the key performance indicator.

**Experiment Details.** All experiments are conducted on public cloud instances with single NVIDIA A800 80 GB GPUs using PyTorch with bfloat16. For multi-GPU results, please refer to Appendix A.3. We enable FlashAttention (Dao et al., 2022) by default to accelerate attention computation.

### 4.2 MAIN RESULTS

**Quantitative Comparison.** Table 1 presents a quantitative evaluation of video generation quality, similarity, and inference speed using VBench. Prompt extension is performed using Qwen2.5 following the instructions from Wan. All main results are evaluated at 480p resolution, while additional VBench scores and similarity results for the Wan model at 720p resolution are provided in Section 4.4.

We adopt two variants to explore different quality-speed trade-offs, controlled by $\delta$, the smaller $\delta$ will switch to rendering model later and get more fidelity results from original sketching model. For TeaCache, we use the open-source implementation and adjust the `l1_distance_thresh` parameter to balance quality and speed. Additionally, we adopt the PAB implementation from HuggingFace Diffusers von Platen et al. (2022), tuning both `block_skip_range` and `timestep_skip_range` to manage the quality-speed trade-off. The baseline models are tuned to ensure they fall within a comparable quality-speed spectrum. Further experimental results, and VBench scores across individual dimensions are provided in Appendix A.4.

For the Wan-based models, SRDiffusion (denoted as SRDiff in the table) achieves significant speedup and consistently outperforms all baselines. Even under the speed-oriented configuration ($\delta = 0.03$), SRDiffusion achieves higher VBench scores than all baselines, even slightly surpassing the original large model Wan 14B, while reducing latency by more than 3×. In terms of similarity metrics, the quality-oriented variant ($\delta = 0.01$) delivers better overall performance. Overall, SRDiffusion demonstrates a clear advantage in acceleration for Wan-based models, with no observable degradation in VBench quality scores.

In the CogVideoX setting, SRDiffusion also demonstrates competitive performance. At $\delta = 0.01$, it nearly matches the original model in VBench score (80.85 vs. 80.89) while achieving the best similarity metrics across all evaluated methods, with a 1.82× speedup. The $\delta = 0.015$ variant offers a slight reduction in quality (VBench 80.51) but achieves a higher speedup of 2.05×, outperforming other baselines with comparable runtime, including PAB and TeaCache-0.15.

The relatively lower acceleration ratio observed for CogVideoX, compared to Wan, is primarily due to the smaller performance gap between the large and small CogVideoX models. Additionally, the switch step in CogVideoX occurs later during inference, which further limits speed gains.

Table 1: Quality results for video generation quality, similarity and inference speed. Similarity metrics are calculated against the original larger model (Wan 14B and CogVideoX 5B) results.

| Method | VBench↑ (Total, Quality, Sematic) | | | Similarity (LPIPS↓, PSNR↑, SSIM↑) | | | Latency | Speedup |
|---|---|---|---|---|---|---|---|---|
| Wan ($832 \times 480$, 5s, $T = 50$) | | | | | | | | |
| Wan 14B | 84.05 | 85.04 | 80.09 | - | - | - | 841 s | $1\times$ |
| Wan 1.3B | 83.12 | 84.28 | 78.50 | - | - | - | 158 s | - |
| $\text{PAB}_{8,[100-970]}$ | 81.83 | 83.14 | 76.59 | 0.247 | 19.94 | 0.706 | 506 s | $1.66\times$ |
| $\text{TeaCache}_{0.2}$ | 83.69 | 84.76 | 79.42 | 0.331 | 16.59 | 0.620 | 427 s | $1.96\times$ |
| **SRDiff** ($\delta = 0.01$) | 84.06 | 84.97 | **80.40** | **0.197** | **20.53** | **0.734** | 296 s | $2.84\times$ |
| **SRDiff** ($\delta = 0.03$) | **84.12** | **85.08** | 80.29 | 0.233 | 19.32 | 0.700 | **262 s** | **$3.21\times$** |
| CogVideoX ($720 \times 480$, 6s, $T = 50$) | | | | | | | | |
| CogVideoX 5B | 80.89 | 82.20 | 75.61 | - | - | - | 213 s | $1\times$ |
| CogVideoX 2B | 80.04 | 81.39 | 74.65 | - | - | - | 75 s | - |
| $\text{PAB}_{8,[100-900]}$ | 76.84 | 78.98 | 68.27 | 0.370 | 17.68 | 0.670 | 105 s | $2.03\times$ |
| $\text{TeaCache}_{0.1}$ | 80.15 | 81.22 | **75.87** | 0.239 | 20.42 | 0.741 | 139 s | $1.53\times$ |
| **SRDiff** ($\delta = 0.01$) | **80.85** | **82.22** | 75.38 | **0.177** | **22.93** | **0.795** | 117 s | $1.82\times$ |
| **SRDiff** ($\delta = 0.015$) | 80.51 | 81.91 | 74.91 | 0.260 | 19.77 | 0.710 | **104 s** | **$2.05\times$** |

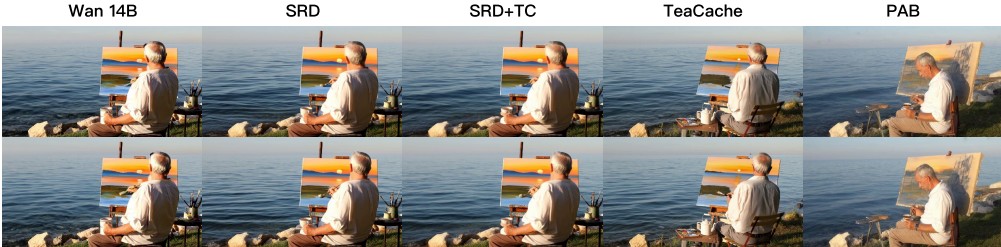

Prompt: An elderly gentleman, with a serene expression, sits at the water's edge, a steaming cup of tea by his side. He is engrossed in his artwork, brush in hand, as he renders an oil painting on a canvas that's propped up against a small, weathered table. The sea breeze whispers through his silver hair, gently billowing his loose-fitting white shirt, while the salty air adds an intangible element to his masterpiece in progress. The scene is one of tranquility and inspiration, with the artist's canvas capturing the vibrant hues of the setting sun reflecting off the tranquil sea.

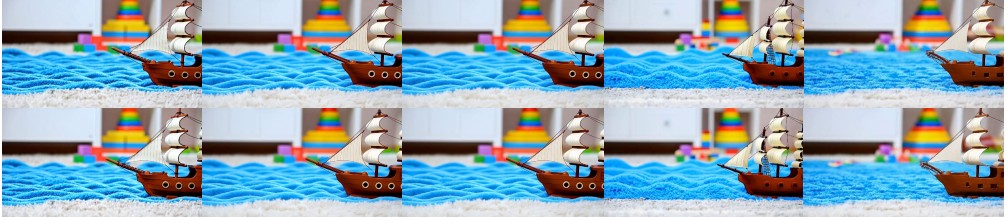

Prompt: A detailed wooden toy ship with intricately carved masts and sails is seen gliding smoothly over a plush, blue carpet that mimics the waves of the sea. The ship's hull is painted a rich brown, with tiny windows. The carpet, soft and textured, provides a perfect backdrop, resembling an oceanic expanse. Surrounding the ship are various other toys and children's items, hinting at a playful environment. The scene captures the innocence and imagination of childhood, with the toy ship's journey symbolizing endless adventures in a whimsical, indoor setting.

Figure 4: Visualization Results. We compare the generation quality between original model, our method and baselines. (SDR: SRDiffusion, TC: TeaCache)

**Visualization Results.** As shown in Figure 4, we present video results generated by the baseline, our method, and the original Wan model for comparison, using selected challenging prompts [2]. The visualizations demonstrate that our method more faithfully preserves the composition of the original model and achieves superior detail generation quality. The "SRD+TC" configuration in the figure illustrates the complementary use of our method with TeaCache, which will be discussed further in Section 4.5. Additional visualization results for Wan and CogVideoX on VBench prompts and more challenging prompts are provided in Appendix A.6.

**User Survey Results.** We randomly sampled 16 prompts from the VBench prompt set and selected an additional 4 challenging prompts from our internal collection. For each prompt, we generated three videos using different models or settings: Original Wan 14B, SRDiffusion ($\delta = 0.01$), and

---

[2] The prompts used for Figure 4 are from `https://github.com/THUDM/CogVideo/blob/main/resources/galary_prompt.md`

TeaCache ($\delta = 0.1$). In the user study, we randomly selected pairs of videos (from the three available versions) for each prompt when each user filling. For each pair, users were asked to compare the two videos and choose A/B has higher quality, or two videos have equal quality. In total, we collected 440 preference choices from 56 participants. Based on these responses, we compute the win rate of each model for every prompt and overall.

| Comparison | Wan 14B v.s. SRDiff | SRDiff v.s. TeaCache | Wan 14B v.s. TeaCache |
|---|---|---|---|
| Overall WinRate | 50.97 | 59.69 | 62.43 |

Table 2: WinRate (%) Results.

### 4.3 CROSS FAMILY COOPERATION

For the cross-family experiments, we employed Wan2.1 14B as the sketching model, and either CogVideoX 5B or Vchitect2.0 2B (Fan et al., 2025) as the rendering model. We set the switching threshold to $\delta = 0.005$, and the results are reported in Table 3. VBench scores across individual dimensions, along with visualization results, are provided in Table 11 and Figure 10 of the Appendix.

Table 3: VBench Score for Cross Family Cooperation.

| VBench Scores | Wan 14B | CogVideoX 5B | Wan+CogVideoX | Vchitect2.0 2B | Wan+Vchitect2.0 |
|---|---|---|---|---|---|
| total score | 84.05 | 80.89 | 81.32 | 78.79 | 81.19 |
| Latency | 841s | 213s | 409s | 253s | 425s |
| semantic score | 80.09 | 75.61 | 77.77 | 76.97 | 80.84 |
| quality score | 85.04 | 82.2 | 82.2 | 79.24 | 81.28 |

### 4.4 SCALING TO HIGHER RESOLUTION

Table 4 presents the VBench and similarity evaluation results for the Wan model at 720p resolution. Since CogVideoX only supports a maximum resolution of 480p, we don't include CogVideoX for evaluation. As shown in the table, SRDiffusion achieves the highest VBench scores across all submetrics and significantly outperforms the baselines in similarity metrics, indicating both higher perceptual and pixel-level fidelity. Moreover, it offers a 2.84× speedup over Wan 14B with much lower latency, demonstrating strong efficiency-quality tradeoffs.

Table 4: VBench and Similariry Results for Wan 720p.

| Method | VBench↑ (Total, Quality, Sematic) | | | Similarity (LPIPS↓, PSNR↑, SSIM↑) | | | Latency | Speedup |
|---|---|---|---|---|---|---|---|---|
| | Wan ($1280{\times}720$, 5s, $T = 50$) | | | | | | | |
| Wan 14B | 83.66 | 84.66 | 79.69 | - | - | - | 3130 s | - |
| Wan 1.3B | 83.57 | 84.69 | 79.06 | - | - | - | 637 s | - |
| TeaCache$_{0.2}$ | 83.45 | 84.51 | 79.22 | 0.331 | 17.04 | 0.668 | 1572 s | 1.99× |
| **SRDiff** ($\delta = 0.01$) | **83.72** | **84.71** | **79.80** | **0.216** | **21.43** | **0.767** | **1104 s** | **2.84×** |

### 4.5 COMPLEMENTARY USE WITH OTHER OPTIMIZATIONS

SRDiffusion operates as a plug-and-play solution and can be seamlessly integrated with various optimization techniques, such as caching mechanisms, system-level enhancements or even step distillation model. In our implementation, we integrate TeaCache with SRDiffusion to further improve efficiency. To ensure stability during the sketching stage, TeaCache is activated only in the rendering stage. We evaluate the combined method, SRDiffusion+TeaCache, on the Wan and CogVideoX using VBench, with the results summarized in Table 5.

As shown, SRDiffusion+TeaCache consistently outperforms both baselines in terms of latency and speedup, while maintaining competitive quality. On the Wan model, it achieves a 3.91× speedup with

the lowest latency (215s), while preserving high semantic and visual fidelity (VBench: 83.82, LPIPS: 0.194, SSIM: 0.741). Similarly, on CogVideoX, SRDiffusion+TeaCache offers the fastest runtime (107s) and the highest speedup (1.99×), with negligible quality trade-offs compared to SRDiffusion alone. These results demonstrate the effectiveness and efficiency of SRDiffusion when combined with other cache-based optimizations. Compared to TeaCache, the main advantage of our method is that we don't skip any steps during the sketching stage, thereby achieving stronger semantic alignment.

Table 5: VBench and Similariry Results for SRDiffusion+TeaCache.

| Method | VBench↑ (Total, Quality, Sematic) | | | Similarity (LPIPS↓, PSNR↑, SSIM↑) | | | Latency | Speedup |
|---|---|---|---|---|---|---|---|---|
| Wan (832×480, 5s, $T = 50$) | | | | | | | | |
| TeaCache$_{0.14}$ | 83.95 | 84.85 | 80.34 | 0.244 | 18.60 | 0.688 | 579 s | 1.45× |
| SRDiff ($\delta = 0.01$) | 84.06 | 84.97 | 80.40 | 0.197 | 20.53 | 0.734 | 296 s | 2.84× |
| **SRDiff+TeaCache** | 83.82 | 84.80 | 79.87 | 0.194 | 20.88 | 0.741 | **215 s** | **3.91×** |
| CogVideoX (720×480, 6s, $T = 50$) | | | | | | | | |
| TeaCache$_{0.1}$ | 80.15 | 81.22 | 75.87 | 0.239 | 20.42 | 0.741 | 139 s | 1.53× |
| SRDiff ($\delta = 0.01$) | 80.85 | 82.22 | 75.38 | 0.177 | 22.93 | 0.795 | 117 s | 1.82× |
| **SRDiff+TeaCache** | 80.24 | 81.48 | 75.28 | 0.187 | 22.56 | 0.791 | **107 s** | **1.99×** |

To further accelerate the process, we incorporate FP8 Attention by adopting SageAttention (Zhang et al., 2024a). The resulting speedup and a sample video (from VBench) are presented in Figure 5. This experiment was conducted on the NVIDIA H20 platform, as SageAttention delivers notable performance improvements only on architectures newer than Hopper. As illustrated, the combination achieves a 6.22× speedup while maintaining consistent semantics and comparable visual quality.

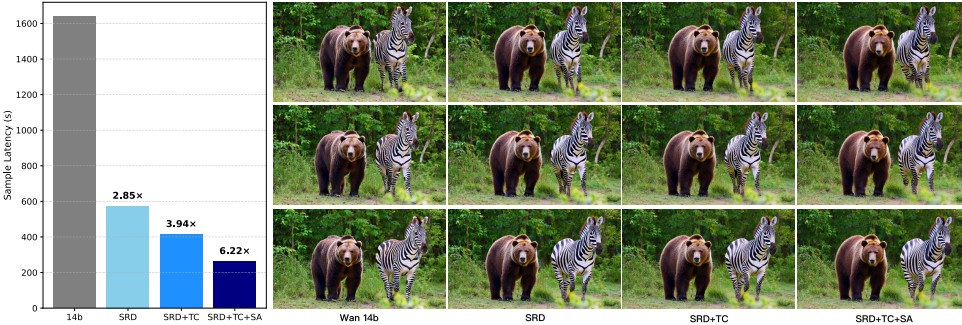

Figure 5: SRDiffusion combined with TeaCache and SageAttention achieves over 6× speedup on a single NVIDIA H20. (SDR: SRDiffusion, TC: TeaCache, SA: SageAttention)

Methods such as TeaCache and PAB, which rely on skip-computing mechanisms, are incompatible with distilled diffusion models. In contrast, SRDiffusion is fully compatible with distillation-based architectures and can deliver additional acceleration with negligible quality degradation. To demonstrate this, we evaluate SRDiffusion on FastWan, a model developed by the FastVideo team that adopts DMD2 (Distribution Matching Distillation) (Yin et al., 2024) to reduce the denoising process to just three inference steps.

In our setup, we use `FastWan2.1-T2V-14B-Diffusers` as the sketching model and `FastWan2.1-T2V-1.3B-Diffusers` as the rendering model, switching to the rendering model after the first timestep. Table 6 presents the corresponding speedup and VBench results.

## 4.6 ANALYSIS OF ADAPTIVE SWITCH

To illustrate the effectiveness of the adaptive switch mechanism, we compare it against a fixed-step switching baseline. Under the $\delta = 0.01$ setting, the average switching step is around 10 for Wan and 15 for CogVideoX. We therefore set these as fixed switching points in the baseline to ensure

Table 6: VBench Results and Speedup for FastWan.

| Method | VBench↑ (Total, Quality, Sematic) | | | Latency | Speedup |
|---|---|---|---|---|---|
| FastWan (832×480, 5s, $T = 3$) | | | | | |
| FastWan2.1-14B | 83.63 | 84.49 | 80.18 | 29.13 s | - |
| FastWan2.1-1.3B | 83.00 | 84.77 | 75.90 | 5.64 s | - |
| **SRDiff** ($ST = 1$) | 83.57 | 84.43 | 80.11 | **13.26 s** | **2.20×** |

equivalent acceleration. The corresponding quality and similarity metrics are reported in Table 7, and the distribution of PSNR values is shown in Figure 6.

Compared to the Fixed-Step Switch, the adaptive switch achieves nearly identical VBench scores but offers a slight advantage in similarity metrics, with the improvement being more pronounced on CogVideoX. By examining the distribution of PSNR values, we observe that adaptive switch results in lower variance, indicating more consistent similarity scores. In challenging cases where similarity is harder to maintain, the adaptive mechanism tends to delay the switch, thereby improving generation quality.

Table 7: Comparison of Adaptive Switch and Fixed-Step Switch.

| Method | VBench↑ (Total, Quality, Sematic) | | | Similarity (LPIPS↓, PSNR↑, SSIM↑) | | |
|---|---|---|---|---|---|---|
| Wan (832×480, 5s, $T = 50$) | | | | | | |
| Fix Step ($T = 10$) | 84.05 | 84.96 | 80.40 | 0.196 | 20.53 | 0.732 |
| Adaptive Switch($\delta = 0.01$) | 84.06 | 84.97 | 80.40 | 0.197 | 20.53 | 0.734 |
| CogVideoX (720×480, 6s, $T = 50$) | | | | | | |
| Fix Step ($T = 15$) | 80.85 | 82.22 | 75.39 | 0.182 | 22.75 | 0.789 |
| Adaptive Switch ($\delta = 0.01$) | 80.85 | 82.22 | 75.38 | 0.177 | 22.93 | 0.795 |

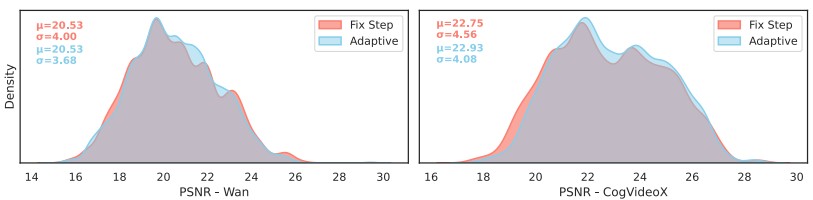

Figure 6: PSNR Distribution for Adaptive Swtich and Fixed-Step Switch.

## 5 CONCLUSIONS

In conclusion, SRDiffusion offers a practical and effective solution to the computational challenges of diffusion-based video generation. By leveraging the semantic strengths of large models during the early, high-noise stages and the efficiency of small models during the later, low-noise stages, SRDiffusion significantly reduces inference time while preserving generation quality. SRDiffusion achieved more than 3× acceleration on Wan without any loss in VBench quality. It also achieved over 2× acceleration on CogVideoX. Additionally, SRDiffusion can be used alongside other methods to achieve even higher acceleration.

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

# A  APPENDIX

## A.1  LLM USAGE

In this paper, LLMs were used to assist in writing, mainly for polishing the text and selecting appropriate words or expressions.

## A.2  ALGORITHM DETAILS OF CROSS-FAMILY COOPERATION

The overall inference procedure for cross-family cooperation is summarized in Algorithm 2. The process begins with the sketching model, which iteratively denoises the latent variable $\mathbf{z}$ from timestep $T$ down to a switching point determined by the condition in Section 3.3. Once the switch condition is satisfied, we jump out from sketching model scheduling loop and calculate a clean sample prediction $\mathbf{x}_0$. This intermediate output is first decoded into pixel space using the sketching model's VAE decoder, ensuring that the intermediate result is expressed in a shared pixel domain. Since the rendering model may require different resolutions and fps, the decoded video $v_s$ is interpolated to match the rendering model's input requirements, producing $v'_s$. The adjusted video is then re-encoded with the rendering model's VAE encoder, yielding a latent representation $x'_0$ that is compatible with the rendering model's latent space. To continue the diffusion process, Gaussian noise corresponding to the target step is added back, forming the new latent $\mathbf{z}$. The precise resumption point $T'$ is determined based on the signal-to-noise ratio (SNR), which provides a principled measure of how much information is preserved in the denoised representation. Finally, the rendering model takes over and carries out the remaining denoising steps until reaching timestep 1, producing the final output $\mathbf{z}_0$.

---

**Algorithm 2** Diffusion Inference Process for Cross-Family Cooperation.

---
1: Initialize latent variable $\mathbf{z}$
2: **for** each timestep $t$ in $\{T, \ldots, 1\}$ **do**
3:     Predict noise: $\hat{\epsilon} \leftarrow \text{SketchingModel}(\mathbf{z}, t)$
4:     Update latents: $\mathbf{z} \leftarrow \text{scheduler.step}(\hat{\epsilon}, t, \mathbf{z})$
5:     **if** switch_condition($\mathbf{z}$) **then**                   ▷ Switch Condition see Sec. 3.3
6:         BREAK
7:     **end if**
8: **end for**
9: Compute clean sample prediction: $\mathbf{x_0} \leftarrow \text{scheduler.compute\_x0}(\hat{\epsilon}, t, \mathbf{z})$
10: Decode to pixel space: $v_s \leftarrow \text{SketchingModel.decode}(\mathbf{x_0})$
11: Align to video parameters of rendering model: $v'_s = \text{interpolate}(v_s)$
12: Re-encode into rendering latent: $x'_0 \leftarrow \text{RenderingModel.encode}(v'_s)$
13: Construct new $\mathbf{z}$ from $x'_0$ with added noise
14: Find $T'$ based on the signal-to-noise ratio.
15: **for** each timestep $t$ in $\{T', \ldots, 1\}$ **do**
16:     Predict noise: $\hat{\epsilon} \leftarrow \text{RenderingModel}(\mathbf{z}, t)$
17:     Update latents: $\mathbf{z} \leftarrow \text{SchedulerStep}(\hat{\epsilon}, t, \mathbf{z})$
18: **end for**
19: **return** Final generated sample $\mathbf{z}_0$

---

## A.3  MULTI-GPU INFERENCE

We also report the speedup achieved under multi-GPU inference. All experiments are conducted on public cloud instances equipped with NVIDIA A800 80GB GPUs connected via NVLink. We adopt Ulysses for parallelizing Wan. Note that for Wan 1.3B, Ulysses is limited to a maximum head size of 4; therefore, for 8 GPUs inference, we combine Ulysses with ring-style sequence parallelism. Table 8 summarizes the multi-GPU inference speedup results for Wan.

## A.4  FURTHER EXPERIMENTAL RESULTS AND VBENCH SCORE FOR EACH DIMENSION

Tables 9 and 10 present the detailed VBench scores of Wan and CogVideoX across various dimensions. These tables also report results for a broader range of parameter configurations than those shown in the Main Results Table 1 in the main text. It can be observed that SRDiffusion demonstrates a clear

| Latency | 1 GPU | 2 GPUs | 4 GPUs | 8 GPUs |
|---|---|---|---|---|
| Wan 14B | 841 s | 486 s | 251 s | 133 s |
| Wan 1.3B | 158s | 96 s | 51 s | 31 s |
| **SRDiff** ($\delta = 0.01$) | 296 s (2.84×) | 175 s (2.78×) | 92 s (2.73×) | 52 s (2.56×) |

Table 8: Latency and Speedup for Multi-GPU Inference.

speed advantage over all baseline configurations. Moreover, its VBench scores are closer to those of larger models, and on Wan model, it even slightly surpasses the 14B model in certain configurations.

Table 9: VBench Score for all dimensions, Wan Model.

| VBench Scores | 14b | 1.3b | $PAB_{C1}$ | $PAB_{C2}$ | $PAB_{C3}$ | $TC_{C1}$ | $TC_{C2}$ | $SRD_{C1}$ | $SRD_{C2}$ | $SRD_{C3}$ | $SRD_{TC}$ |
|---|---|---|---|---|---|---|---|---|---|---|---|
| **total score** | 84.05 | 83.12 | 83.13 | 82.42 | 81.83 | 83.95 | 83.69 | 83.87 | 84.06 | 84.12 | 83.82 |
| **speedup** | 1× | - | 1.04× | 1.66× | 1.81× | 1.45× | 1.96× | 2.23× | 2.84× | 3.21× | 3.91× |
| **semantic score** | 80.09 | 78.5 | 79.70 | 78.16 | 76.59 | 80.34 | 79.42 | 79.85 | 80.4 | 80.29 | 79.87 |
| object class | 93.09 | 90.87 | 92.52 | 90.47 | 88.58 | 93.01 | 91.04 | 92.47 | 92.44 | 92.1 | 92.5 |
| multiple objects | 81.16 | 77.26 | 79.79 | 77.04 | 69.59 | 80.79 | 78.32 | 80.37 | 80.93 | 81.08 | 81.2 |
| human action | 98 | 95.2 | 97.80 | 97.6 | 96.40 | 97.4 | 97 | 97.80 | 97.8 | 97.8 | 97.2 |
| color | 84.6 | 86.25 | 84.43 | 83.67 | 83.61 | 83.63 | 85.88 | 84.03 | 85.82 | 85.1 | 84.8 |
| spatial relationship | 78.65 | 76.66 | 76.53 | 74.42 | 72.56 | 80.32 | 77.28 | 78.73 | 79.66 | 79.53 | 78.86 |
| scene | 66.58 | 66.97 | 67.72 | 65.06 | 65.17 | 69.81 | 66.65 | 66.42 | 68.23 | 68.18 | 66.61 |
| appearance style | 77.2 | 73.3 | 77.02 | 76.18 | 76.39 | 76.81 | 77.34 | 76.74 | 76.49 | 76.35 | 76 |
| temporal style | 68.35 | 67.75 | 68.35 | 66.9 | 65.80 | 68.21 | 67.83 | 68.71 | 68.76 | 68.87 | 68.46 |
| overall consist | 73.16 | 72.25 | 73.13 | 72.12 | 71.18 | 73.05 | 73.41 | 73.35 | 73.43 | 73.57 | 73.19 |
| **quality score** | 85.04 | 84.28 | 83.98 | 83.49 | 83.14 | 84.85 | 84.76 | 84.87 | 84.97 | 85.08 | 84.8 |
| subject consist | 93.91 | 93.44 | 93.79 | 93.53 | 93.22 | 93.84 | 93.75 | 93.94 | 94 | 93.94 | 93.97 |
| background consist | 97.06 | 96.55 | 95.00 | 94.34 | 94.14 | 97.08 | 97.09 | 96.99 | 97.01 | 97.06 | 96.98 |
| temporal flickering | 96.84 | 97.63 | 97.14 | 97.11 | 97.17 | 96.82 | 97.03 | 96.79 | 96.79 | 96.79 | 96.68 |
| motion smoothness | 93.76 | 92.9 | 94.10 | 94.72 | 94.82 | 93.86 | 93.89 | 93.69 | 93.76 | 93.79 | 94 |
| dynamic degree | 37.5 | 37.36 | 35.97 | 35.28 | 35.00 | 36.95 | 37.09 | 37.22 | 37.78 | 38.61 | 37.22 |
| aesthetic quality | 66.9 | 64.35 | 63.16 | 62.05 | 61.34 | 66.75 | 66.54 | 66.28 | 66.22 | 66.17 | 66.08 |
| imaging quality | 66.76 | 65.57 | 66.72 | 65.64 | 64.72 | 66.22 | 65.53 | 66.76 | 66.77 | 66.64 | 66.3 |

Table 10: VBench Score for all dimensions, CogVideoX Model.

| VBench Scores | 5b | 2b | $PAB_{C1}$ | $PAB_{C2}$ | $TC_{C1}$ | $TC_{C2}$ | $TC_{C3}$ | $SRD_{C1}$ | $SRD_{C2}$ | $SRD_{C3}$ | $SRD_{C4}$ | $SRD_{TC}$ |
|---|---|---|---|---|---|---|---|---|---|---|---|---|
| **total score** | 80.89 | 80.04 | 80.68 | 76.84 | 80.15 | 79.16 | 78.15 | 80.83 | 80.85 | 80.51 | 80.12 | 80.24 |
| **speedup** | 1× | - | 1.36× | 2.03× | 1.53× | 2.01× | 2.39× | 1.65× | 1.82× | 2.05× | 2.32× | 1.99× |
| **semantic score** | 75.61 | 74.65 | 75.13 | 68.27 | 75.87 | 74.09 | 73.59 | 75.36 | 75.38 | 74.91 | 74.65 | 75.28 |
| object class | 87.41 | 85.9 | 86.33 | 77.17 | 86.16 | 84.03 | 82.74 | 87.69 | 86.69 | 87.1 | 87.36 | 87.37 |
| multiple objects | 65.09 | 65.38 | 65.26 | 45.85 | 66.01 | 58.78 | 58.84 | 66.36 | 67.96 | 66.17 | 66.07 | 65.11 |
| human action | 98.2 | 96.8 | 98 | 93 | 97.8 | 97 | 96.8 | 97.40 | 97 | 97 | 96.6 | 97.2 |
| color | 86.84 | 86.12 | 86.95 | 85.63 | 87.16 | 84.72 | 85.24 | 87.82 | 88.16 | 87.07 | 85.1 | 87.66 |
| spatial relationship | 55.03 | 54.87 | 53.8 | 42.97 | 57.6 | 58.77 | 56.69 | 54.73 | 54.23 | 54.23 | 54.54 | 54.78 |
| scene | 66.46 | 63.78 | 64.91 | 55.87 | 65.6 | 61.59 | 61.04 | 63.46 | 63.71 | 62.28 | 61.96 | 63.73 |
| appearance style | 81.13 | 81.87 | 80.96 | 80.39 | 81.66 | 81.83 | 81.76 | 80.85 | 81.13 | 81.41 | 81.8 | 82.01 |
| temporal style | 66.87 | 64.84 | 66.48 | 62.55 | 66.62 | 66.43 | 65.58 | 66.57 | 66.48 | 66.24 | 65.77 | 66.1 |
| overall consist | 73.46 | 72.25 | 73.46 | 70.99 | 74.18 | 73.68 | 73.63 | 73.35 | 73.08 | 72.66 | 72.66 | 73.6 |
| **quality score** | 82.2 | 81.39 | 82.07 | 78.98 | 81.22 | 80.43 | 79.28 | 82.19 | 82.22 | 81.91 | 81.48 | 81.48 |
| subject consist | 92.55 | 92.81 | 92.49 | 90.72 | 91.92 | 91.41 | 90.97 | 92.49 | 92.28 | 92.11 | 91.85 | 91.8 |
| background consist | 94.48 | 94.19 | 94.42 | 93.97 | 94.07 | 94.06 | 94.14 | 94.34 | 94.23 | 94.04 | 93.64 | 93.87 |
| temporal flickering | 93.9 | 93.58 | 94.2 | 95.47 | 93.66 | 93.93 | 93.88 | 93.74 | 93.34 | 93.01 | 92.58 | 93.44 |
| motion smoothness | 92.8 | 93.04 | 93.07 | 93.62 | 91.9 | 91.6 | 90.87 | 92.62 | 91.94 | 91.25 | 90.39 | 91.56 |
| dynamic degree | 38.47 | 36.07 | 38.05 | 30 | 35.84 | 33.61 | 29.03 | 38.61 | 39.45 | 38.89 | 38.89 | 37.91 |
| aesthetic quality | 60.69 | 59.19 | 60.78 | 57.09 | 59.95 | 58.78 | 57.95 | 60.66 | 60.59 | 60.4 | 59.89 | 59.79 |
| imaging quality | 61.44 | 60.15 | 60.45 | 52.5 | 60.6 | 59.42 | 58.51 | 61.79 | 62.59 | 62.7 | 62.39 | 61.21 |

**Configure Details for Wan.** C1 in the PAB represents the default configuration, while C2 and C3 apply `block_skip_range=8` with `timestep_skip_range` set to $[100, 950]$ and $[100, 970]$, respectively. For TeaCache (denoted as TC in the table), configurations C1 and C2 correspond to `l1_distance_thresh` values of 0.14 and 0.2. For SRDiffusion (denoted as SRD in the table), configurations C1, C2, and C3 use $\delta$ values of 0.002, 0.01, and 0.03, respectively. The $SRD_{TC}$ configuration uses $\delta = 0.01$ in combination with TeaCache with a threshold of 0.14.

**Configure Details for CogVideoX.** C1 in the PAB represents the default configuration, while C2 apply `block_skip_range=8` with `timestep_skip_range=[100,900]`. For TeaCache (denoted as TC in the table), configurations C1, C2, C3 correspond to `l1_distance_thresh values` of 0.1, 0.15 and 0.2. For SRDiffusion (denoted as SRD in the table), configurations C1, C2, C3 and C4 use $\delta$ values of 0.008, 0.01, 0.015 and 0.03, respectively. The $SRD_{TC}$ configuration uses $\delta = 0.01$ in combination with TeaCache with a threshold of 0.1.

Tables 11 present the detailed VBench scores of cross family experiments across various dimensions. We employ Wan 2.1 14B as the sketching model, while CogVideoX 5B and Vchitect2.0 2B serve as the rendering models, with $\delta = 0.005$ applied in both settings. We observe that SRDiffusion can still be performed under the cross-family condition. Using Wan 14B as the sketching model substantially improves the semantic score, with notable gains in dimensions such as *object class*, *multiple objects*, and *spatial relationships*. However, due to the limited generative capacity of the smaller rendering models, the final SRDiffusion scores lie between the sketching and rendering models.

Table 11: VBench Score for all dimensions, Cross Family Cooperation.

| VBench Scores | Wan 14B | CogVideoX 5B | Wan+CogVideoX | Vchitect2.0 2B | Wan+Vchitect2.0 |
|---|---|---|---|---|---|
| **total score** | 84.05 | 80.89 | 81.32 | 78.79 | 81.19 |
| **semantic score** | 80.09 | 75.61 | 77.77 | 76.97 | 80.84 |
| object class | 93.09 | 87.41 | 91.14 | 90.24 | 94.51 |
| multiple objects | 81.16 | 65.09 | 74.73 | 71.17 | 82.96 |
| human action | 98 | 98.2 | 97.8 | 96.40 | 98 |
| color | 84.6 | 86.84 | 83.08 | 85.61 | 81.48 |
| spatial relationship | 78.65 | 55.03 | 65.64 | 56.81 | 77.33 |
| scene | 66.58 | 66.46 | 67.76 | 65.22 | 68.41 |
| appearance style | 77.2 | 81.13 | 79.55 | 85.42 | 81.03 |
| temporal style | 68.35 | 66.87 | 66.86 | 67.75 | 69.70 |
| overall consist | 73.16 | 73.46 | 73.57 | 74.12 | 74.18 |
| **quality score** | 85.04 | 82.2 | 82.2 | 79.24 | 81.28 |
| subject consist | 93.91 | 92.55 | 93.56 | 90.64 | 92.35 |
| background consist | 97.06 | 94.48 | 95.49 | 94.01 | 95.38 |
| temporal flickering | 96.84 | 93.9 | 96.28 | 88.45 | 95.33 |
| motion smoothness | 93.76 | 92.8 | 95.06 | 87.07 | 90.33 |
| dynamic degree | 37.5 | 38.47 | 26.25 | 26.53 | 26.95 |
| aesthetic quality | 66.9 | 60.69 | 61.9 | 62.59 | 62.70 |
| imaging quality | 66.76 | 61.44 | 65.84 | 65.78 | 65.29 |

## A.5 DISTRIBUTION FOR ADAPTIVE SWITCH STEPS

We analyzed the step distribution under different $\delta$ values using the standard prompt set of VBench, focusing on Wan 480p, Wan 720p, and CogVideoX 480p. Box plots were used to visualize the data. We observed that a larger $\delta$ leads to earlier switching, resulting in a higher acceleration ratio, while a smaller $\delta$ causes later switching, thereby staying more faithful to the original output of the Sketching Model. Notably, for different prompts, the switching step range of Wan is significantly wider than that of CogVideoX, which is consistent with our observations during the design of the evaluation metric, referring Figure 3.

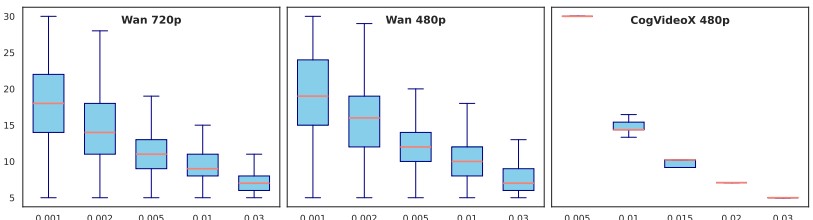

Figure 7: Distribution for Adaptive Switch Steps.

## A.6 MORE VISUALIZATION RESULTS

In this section, we present several visual examples of video generation results:

**Figures 8 and 9 showcase randomly selected samples generated by Wan and CogVideoX on prompts from VBench.** For both the Wan and CogVideoX models, the configurations used for visualization include $SRD_{C2}$, $SRD_{TC}$, $SRD_{TC}$ with SageAttention, $PAB_{C2}$, and $TC_{C1}$. Detailed parameter settings can be found in Appendix A.4. Compared to PAB and TeaCache, SRDiffusion demonstrates a better ability to follow the original model's generation. It produces noticeably more consistent results in terms of composition and motion.

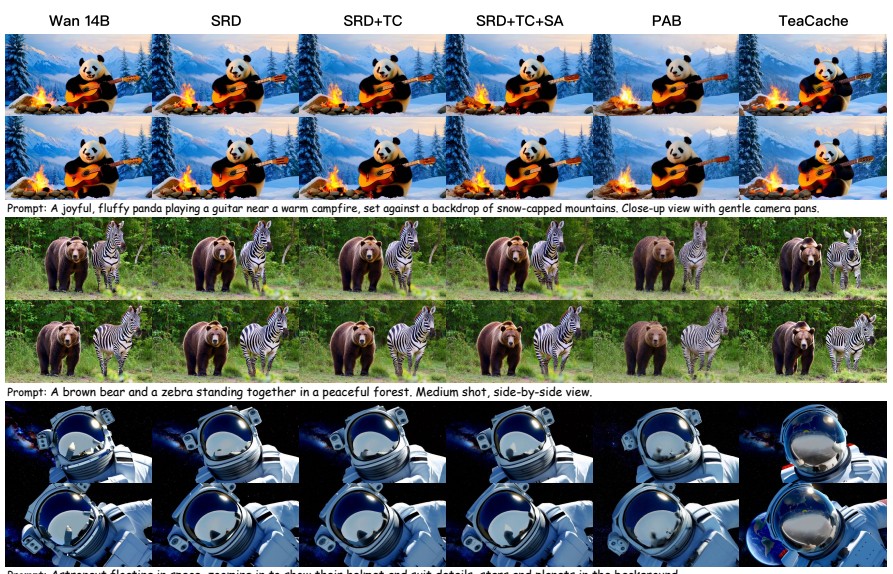

Figure 8: Visualization Results of Wan model for VBench Prompts. (SDR: SRDiffusion, TC: TeaCache, SA: SageAttention)

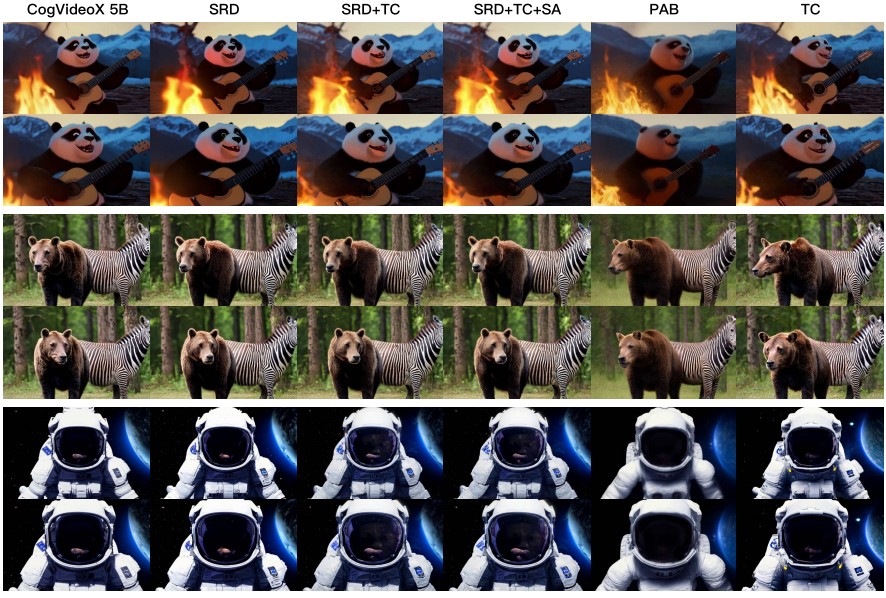

Figure 9: Visualization Results of CogVideoX model for VBench Prompts, prompt same from Figure 8. (SDR: SRDiffusion, TC: TeaCache, SA: SageAttention)

**Figure 10 shows the visualization of cross family cooperation on prompts from VBench.** We use Wan 2.1 14B as the sketching model and use CogVideoX 5B and Vchitect2.0 2B as the renderning

model. We employ Wan 2.1 14B as the sketching model, while CogVideoX 5B and Vchitect2.0 2B serve as the rendering models. As shown, the generated videos follow the composition and motion guided by the sketching model, thereby enhancing semantic ability.

**Figure 11 illustrates the outputs of Wan and CogVideoX on a set of challenge prompts.** The prompts in VBench are relatively simple, so we collected more challenging prompts from the open-source community for further evaluation. In this part, Wan was evaluated at 720p resolution, while CogVideoX was tested at 480p (as it only supports 480p). We observed that, compared to PAB and TeaCache, SRDiffusion better preserves the generation quality of the original model. For example, in the first prompt, both the sails and the background are more accurately rendered. In the second prompt, the structure of the house and the position of the picture frame are more similar. In the third prompt, the elderly man's appearance and the subject of his painting are better consistency. In the fourth prompt, the style of the boat and the background are more faithfully depicted.

On these more challenging prompts, SRDiffusion demonstrates a stronger ability to follow the original model's generation while adhering more closely to the given instructions and producing finer details. This improved detail generation may stem from the fact that, unlike the baseline methods which skip certain computations, SRDiffusion employs a small rendering model that retains more reliable detail generation capabilities.

**Figure 12 displays video generation results from SRDiffusion under different values of the parameter $\delta$.** For Wan, we visualized the results under three different settings: $\delta = 0.002$, $0.01$, and $0.03$. For CogVideoX, we tested three values as well: $\delta = 0.008$, $0.01$, and $0.015$. We observe that smaller $\delta$ values tend to more closely follow the outputs generated by the large model. For example, in the second prompt of Wan, the style of the sunglasses differs when $\delta = 0.03$. However, it's worth noting that in most cases, such differences do not indicate a decline in generation quality.

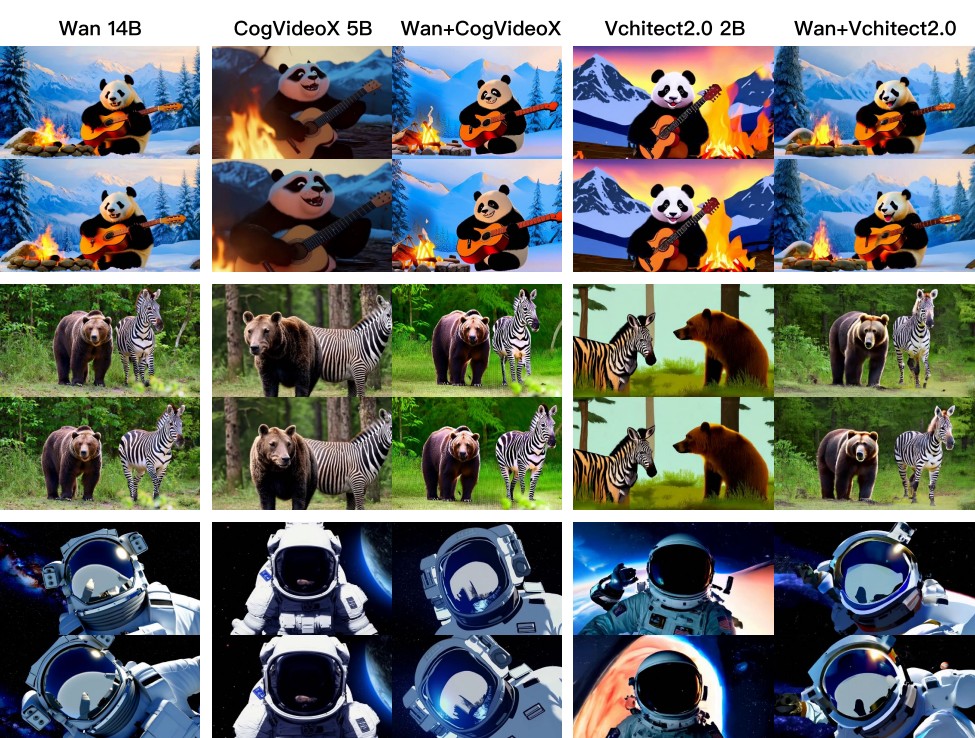

Figure 10: Visualization Results of Cross Family Cooperation for VBench prompt, prompt same from Figure 8.

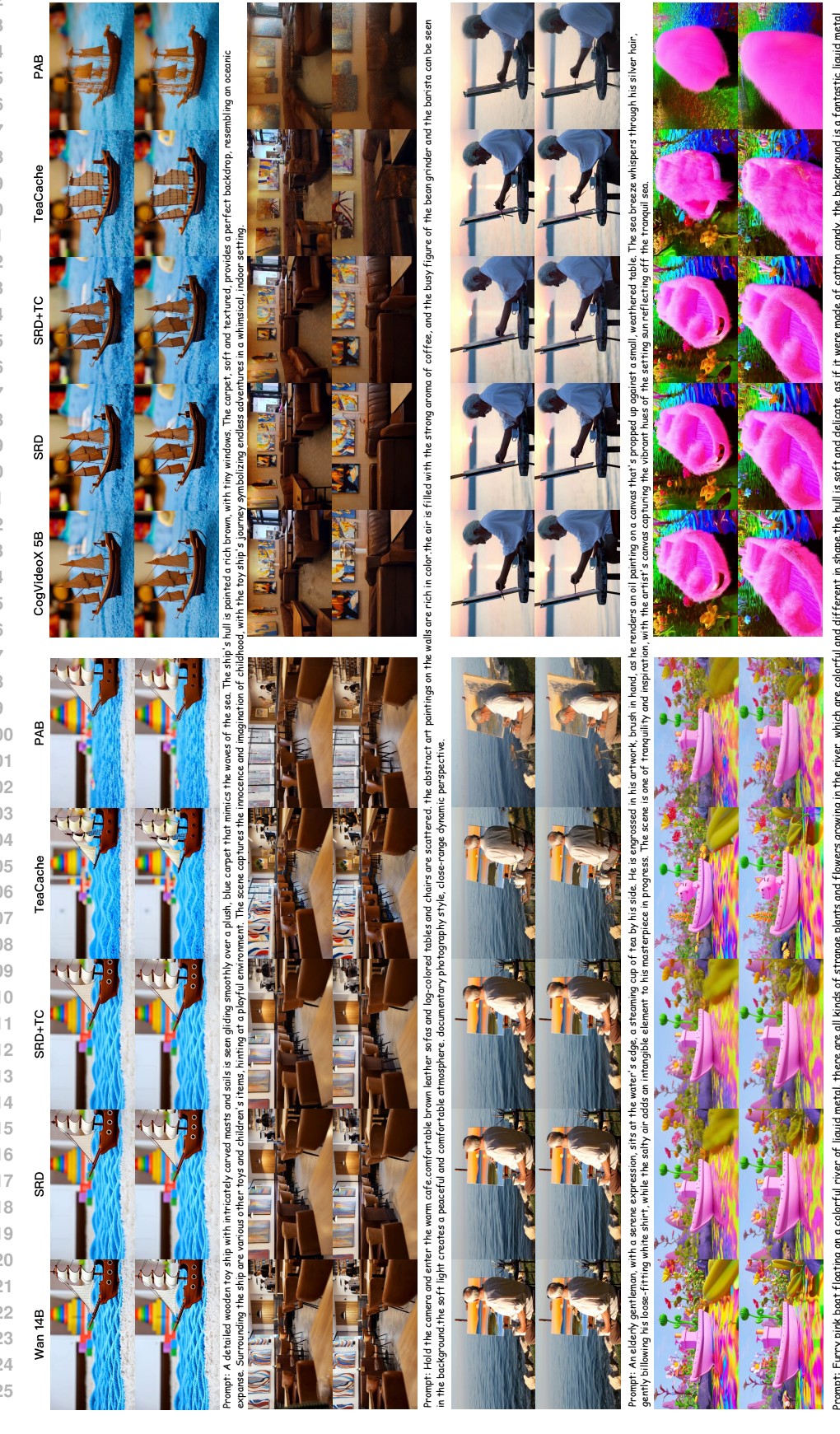

Figure 11: Visualization Results of Wan and CogVideoX model for Challenging Prompts. (SDR: SRDiffusion, TC: TeaCache, SA: SageAttention)

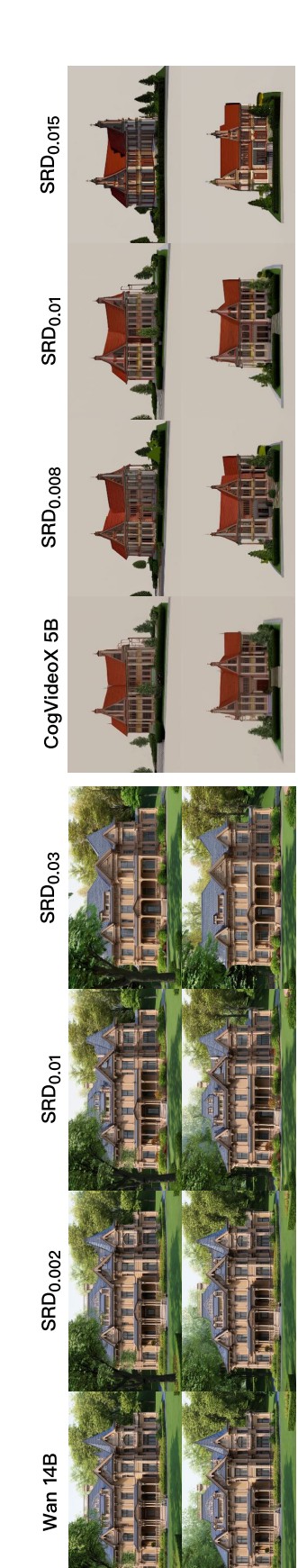

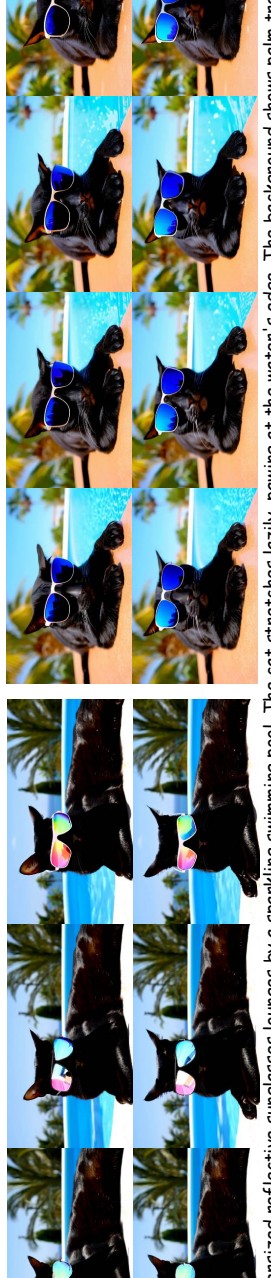

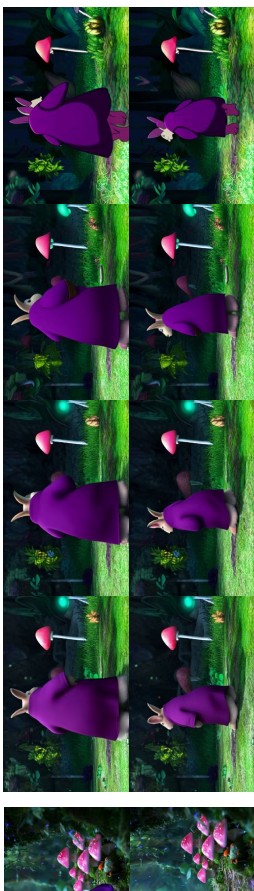

Figure 12: Comparison of visualization results from SRDiffusion using various $\delta$ values and the original model output. (SDR: SRDiffusion)

