# OpenReview forum: "SRDiffusion: Accelerate Diffusion Inference via Sketching-Rendering Cooperation"
_ICLR.cc/2026/Conference — Submitted to ICLR 2026_

### Official Review · Reviewer_gjjr · 2025-10-14

**Soundness:** 2
**Presentation:** 2
**Contribution:** 2
**Rating:** 2
**Confidence:** 4

**Summary:**

This paper presents a training-free algorithm to accelerate the sampling process. The core idea is to use a large model for the initial denoising steps and then switch to a smaller model for the later stages. This reduces inference time compared to exclusively using the large model.

However, the work's primary weakness is its simplicity and lack of novelty. The contributions are twofold: (1) proposing to substitute the large model with a small one in the later phase of denoising, and (2) providing a metric to determine when to make this switch. It is important to note that the proposed metric is not original, as a similar concept has already been discussed in Teacache.

**Strengths:**

The article's strength lies in its well-written and easy-to-understand content, but that is where its merits end.

**Weaknesses:**

1. Limited Novelty and Practicality: The core idea of substituting a large model with a smaller one during the denoising process is overly simplistic and lacks sufficient novelty for a publication at this venue. Moreover, the method's practical applicability is questionable. It relies on the pre-existence of an official, smaller version of the large model. The paper fails to address how the method would be applied to models without a corresponding small variant, such as HunyuanVideo. The authors should clarify whether a new small model must be trained, and how that would impact the "training-free" claim of the algorithm.

2. Unconvincing Motivation: I have reservations about the paper's central motivation. The authors justify their approach by citing VBench results, claiming that the quality gap between the Wan 14B and 1.3B models is negligible. This premise is weak for two reasons: VBench is not a widely accepted authoritative benchmark, and its metrics may not reliably capture perceptual quality. The claim contradicts the general consensus in the community that the visual quality of WanX-14B is significantly superior to WanX-1.3B, a difference that is readily apparent to the human eye. Relying on a questionable benchmark that conflicts with perceptual reality undermines the paper's foundation.

3. Incomplete Survey of Related Work: The related work section is missing a crucial category of acceleration techniques: step distillation. Methods like DMD2, used in works such as FastVideo [1], achieve substantial acceleration (20-30x) and are highly relevant. The authors should include a thorough discussion of these methods and position their work in relation to them.

4. Redundant Writing: The manuscript is repetitive. The core methodology is restated multiple times throughout the paper without offering new insights or details, which detracts from its clarity and impact.

[1]. FastVideo: A Unified Framework for Accelerated Video Generation, https://github.com/hao-ai-lab/FastVideo

**Questions:**

No

---

> ### Author Response · Authors · 2025-11-20
>
> We would like to thank the Reviewer for the constructive feedback. Please find our responses below:
> 1. **1)** SRDiffusion also supports cross-family cooperation, as demonstrated in Section 4.3 with examples such as Wan+CogVideo and Wan+Vchitect2.0. In addition, HunyuanVideo can be used as either the sketching or rendering model within the cooperation pipeline. In this sense, SRDiffusion is able to flexibly select models from a model zoo to construct the pipeline, further reinforcing our claim that the method is training-free. **2)** The core idea may appear simple, the design, analysis, and validation of SRDiffusion address multiple technical challenges. The proposed framework introduces several innovations: adaptive model switching, cross-family cooperation, and compatibility with existing acceleration techniques such as caching and distillation. And our extensive experiments, including integration with FastWan, high-resolution VBench evaluations, and user studies, demonstrate that SRDiffusion achieves substantial acceleration while maintaining semantic and perceptual quality.
> * **1)** Our motivation arises from the observation that large models exhibit clear advantages in semantic understanding compared with smaller models, while their superiority in visual fidelity is often less evident. This trend appears not only in the two VBench dimensions (semantics and quality), but is also consistent with our direct inspection of generated videos. We consider VBench’s semantic evaluations reliable: across dimensions such as object classification, multi-object scenes, and spatial relationships, although VBench relies on learned models for scoring, the metrics consistently reflect the performance gap between large and small models. Moreover, we find that the videos produced by SRDiffusion are nearly indistinguishable from those generated by the large model in terms of semantic content. The remaining differences are mostly minor visual details, for example, the exact style of a cup or the pattern on a hot-air balloon, which do not indicate degraded quality. These variations are subjective in nature, and we cannot judge them as clearly better or worse. **2)** VBench is a widely adopted evaluation suite for research on video generation and acceleration, and many recent studies rely on its standardized settings. Following the common evaluation protocol used in recent video-diffusion acceleration work (e.g., TeaCache, PAB, AdaCache, FasterCache, and MagCache), we report VBench results, including similarity metrics and visualizations. **3)** We also include a user study in Section 4.2 to further validate the effectiveness of our method. We observe that the win rate of Wan 14B over SRDiffusion is around 50%, indicating that the quality degradation introduced by SRDiffusion is nearly imperceptible to human evaluators. Additionally, the win rate of SRDiffusion over TeaCache demonstrates that our method achieves superior perceptual quality compared with TeaCache.
> * We have added SRDiffusion support to FastWan from the FastVideo team, and the corresponding experimental results are presented in Section 4.5. Both FastWan2.1-14B and FastWan2.1-1.3B use DMD, whose denoising process consists of three steps. Accordingly, in SRDiffusion, we run the first step with FastWan2.1-14B and switch to FastWan2.1-1.3B for the remaining steps, achieving a 2.20× speedup while preserving performance, especially in terms of semantic quality.
> * We have revised the references, improved the writing to reduce redundancy, and added additional related work, including studies on distillation.

---

### Official Review · Reviewer_j72M · 2025-10-29

**Soundness:** 3
**Presentation:** 3
**Contribution:** 2
**Rating:** 4
**Confidence:** 4

**Summary:**

This paper introduces a training-free acceleration method for video diffusion sampling, called SRDiff, which leverages collaboration between large and small models to reduce the inference computational cost. The main idea behind the method is like the speculative decoding. From the experiments on VBench, RSDiff can accelerate the sampling speed up to 3x and 2x on Wan and CogVideoX, respectively.

**Strengths:**

The core idea of this paper is initutive and straightforwad. It utilizes the large model handles high-noise steps to ensure semantic and motion fidelity, while utilizes the smaller model refines visual details in low-noise steps. The authors provide the solutions on the same VAE family and cross-frame VAE family.

**Weaknesses:**

First, I need to point out the writing issues of this paper. The reference format is totally wrong. I think the authors mis-use \citep and \citet, which makes the paper hard to read. And the core method are introduced several times during the whole paper. I hope the authors can provide more interesting phonemenon or insights from this method.

Next, I need to point out the technical issues.

1. Insufficient related work.  As a training-free acceleration method, this paper misses a lot of disscussions with other acceleration methods like FastVideo [1], CausVid [2] etc.

2. Limited novelty. As I mentioned in before, the idea of this method is initutive and straightforward, but this method has been applied on several models like $\textbf{Wan2.2}$ [3], which utilizes a MoE structure with a high-denoise model and low-denoise model, and $\textbf{SDXL-refiner}$ [4], an ensemble of experts pipeline.

3. Insufficient experiment results. Through the whole paper, all experiments are conducted on the VBench.

4. The method utilizes two models for a single video generation, how about the memory cost of SRDiff?

[1] FastVideo: https://github.com/hao-ai-lab/FastVideo

[2] From Slow Bidirectional to Fast Autoregressive Video Diffusion Models: https://github.com/tianweiy/CausVid

[3] Wan2.2: https://github.com/Wan-Video/Wan2.2

[4] SDXL-refiner: https://huggingface.co/stabilityai/stable-diffusion-xl-refiner-1.0

**Questions:**

I think the authors need to clarify the contributions and the significance of this paper. Provide more insights behind the proposed methods. More details please see the weaknesses.

---

> ### Author Response · Authors · 2025-11-20
>
> We would like to thank the Reviewer for the constructive feedback. Please find our responses below:
> * We have revised the references, improved the writing to reduce redundancy, and added additional related work, including studies on distillation.
> * These works pursue different objectives. SRDiffusion focuses on acceleration, whereas Wan2.2 uses two separate models and switches between them at fixed steps, which does not provide any speedup. Its goal is to enhance generation quality, and it is trained with varying noise levels to support such switching. Similarly, SDXL-Refine functions more like an img2img-style refinement stage: it improves visual quality through an additional post-processing pass, but at the cost of increased generation time. In contrast, SRDiffusion performs adaptive switching and can be combined with many other optimization techniques, including cache-based methods, quantization, and distilled models. Moreover, SRDiffusion provides comprehensive experimental evidence to validate these advantages.
> * **1)** We provide VBench evaluations, similarity metrics, and visualizations following the evaluation protocol commonly adopted by recent video-diffusion acceleration work such as TeaCache, PAB, AdaCache, FasterCache, and MagCache. **2)** We also include user study results in the end of Section 4.2. We find that the win rate of Wan 14B over SRDiffusion is around 50%, indicating that the quality degradation introduced by SRDiffusion is nearly imperceptible to human evaluators. Furthermore, the win rate of SRDiffusion over TeaCache shows that our method achieves better perceptual quality than TeaCache.
> * In our SRDiffusion experiments, an 80GB GPU can hold both Wan 14B and Wan 1.3B simultaneously, enabling switching between them. For more resource-constrained devices, we can adopt a Sketching–Rendering Disaggregation design, similar to PD Disaggregation, where different types of GPUs handle different stages. For example, the sketching stage can be executed on high-end GPUs, while the rendering stage can be run on GPUs with smaller memory capacity or even on edge devices.

---

> > ### Comment · Reviewer_j72M · 2025-11-26
> > **Response to the authors**
> >
> > Thanks for the author's response. I have read the rebuttal carefully, and I am willing to raise my score.

---

### Official Review · Reviewer_mRpW · 2025-10-31

**Soundness:** 3
**Presentation:** 3
**Contribution:** 2
**Rating:** 4
**Confidence:** 4

**Summary:**

The paper proposes SRDiffusion, a training-free method to accelerate video diffusion inference by combining a large model for early high-noise steps (semantic fidelity) with a smaller model for later low-noise steps (detail refinement). An adaptive switching metric ensures a good balance between speed and quality. Experiments on Wan and CogVideoX show up to 3x speedup with negligible quality loss, outperforming caching-based baselines and complementing other optimizations.

**Strengths:**

1) The paper is well written; it is easy to read and understand.
2) The core idea is very simple and shows good performance as evidenced by quality metrics.
3) The method is training-free, does not require any additional training or finetuning.
4) The approach is flexible and extendable to different video diffusion transformer model families; cross family experiments are especially interesting.
5) The idea is orthogonal to other acceleration techniques, so it can be combined with other methods to improve efficiency even further.

**Weaknesses:**

1) While effective and new, the core concept (switching between large and small models) is relatively simple and may be seen as incremental rather than fundamentally novel.
2) The paper relies solely on automated metrics (VBench, LPIPS, SSIM, PSNR), which are imperfect proxies for perceptual quality; lack of user studies or preference tests is concerning for a video generation task.
3) No video examples in supplementary: Only rolled-out frames are shown, making it hard to judge temporal consistency and overall perceptual quality.
4) Maintaining two or more models during inference may pose memory and deployment challenges, especially in resource-constrained environments; this is not addressed in detail.

**Questions:**

1) How sensitive is the delta threshold to different prompt types? Could a learned policy outperform the current heuristic?
2) Have you considered latent alignment techniques (e.g., learned mapping) to reduce quality loss when switching across different VAEs?
3) What are the memory and compute implications of maintaining two models during inference? Any strategies for deployment on resource-constrained devices?
4) Why were no user studies or preference tests included? Do you plan to validate perceptual quality beyond automated metrics?
5) Will SRDiffusion work for step-distilled models (e.g., DMD) that achieve high-quality generation in 1–4 steps? If not, doesn’t this severely limit the practical relevance of your approach compared to these highly efficient alternatives?

---

> ### Author Response · Authors · 2025-11-20
>
> We would like to thank the Reviewer for the constructive feedback. Please find our responses below:
> * We have uploaded 20 prompt-based video generation samples to Google Drive for your reference. (https://drive.google.com/drive/folders/1zyKouSuulfkwtRT9pNx_u_MJLCLnY4D6?usp=sharing)
> * **1) Memory.** An 80GB GPU is sufficient to load both Wan 14B and Wan 1.3B simultaneously, enabling switching between them. For more resource-constrained devices, we can adopt a Sketching–Rendering Disaggregation design (similar to PD Disaggregation), where different types of GPUs handle different stages. For example, the sketching stage can run on high-end GPUs, while the rendering stage can run on GPUs with smaller memory or even on edge devices. **2) Computation.** If switching does not involve changing VAEs, the additional overhead is almost negligible. If switching across different VAEs is required, our method introduces only one extra sketching-VAE decode and one rendering-VAE encode. The results in Table 3 already include these costs and still show substantial speedup.
> * We include user study results in the end of Section 4.2. We find that the win rate of Wan 14B over SRDiffusion is around 50%, indicating that the quality degradation introduced by SRDiffusion is nearly imperceptible to human evaluators. Furthermore, the win rate of SRDiffusion over TeaCache shows that our method achieves better perceptual quality than TeaCache.
> * We have added SRDiffusion support to FastWan from the FastVideo team, and the corresponding experimental results are presented in Section 4.5. Both FastWan2.1-14B and FastWan2.1-1.3B use DMD2, whose denoising process consists of three steps. Accordingly, in SRDiffusion, we run the first step with FastWan2.1-14B and switch to FastWan2.1-1.3B for the remaining steps, achieving a 2.20× speedup while preserving performance, especially in terms of semantic quality.
> * We provide the distribution of switch steps under different δ values in Appendix A.6. We observe that Wan is more sensitive to different prompts, while CogVideo is much less sensitive. Consistent with our observations in Figure 3, CogVideo’s Dt curves remain similar across prompts, whereas Wan’s curves vary significantly. We believe this is not a single-objective optimization problem: there is an inherent trade-off between generation quality and generation speed, which necessitates a threshold. Nonetheless, exploring learnable switching strategies remains an interesting direction for future work.
> * In our current cross-family cooperation setup, we choose to transfer information at the pixel level. While this approach incurs some computational overhead, it remains feasible. A learned mapping, however, would require retraining for every possible combination of sketching and rendering models, which would significantly reduce the practicality of the method in real-world applications. Therefore, we opt to keep the method training-free to maximize its applicability and feasibility in practice.

---

### Official Review · Reviewer_6FUF · 2025-11-01

**Soundness:** 3
**Presentation:** 3
**Contribution:** 3
**Rating:** 6
**Confidence:** 3

**Summary:**

The paper observes large models excel at semantics in high-noise steps, while small models are efficient at low-noise steps. SRDiffusion therefore uses a large model for early steps (“Sketching”) and a small model for later steps (“Rendering”), with an adaptive switch metric. Standard diffusion notation is used: forward process $x_t=\sqrt{\alpha_t},x_{t-1} + \sqrt{1-\alpha_t},\varepsilon_t$ with $\varepsilon_t!\sim!\mathcal{N}(0,I)$, and decoding via a scheduler $\Phi$ to compute $x_{t-1}=\Phi!\big(x_t,t,O(x_t,t)\big)$. The switch metric is $D_t=\tanh!\big(\lVert x_t-x_{t+1}\rVert_1/\lVert x_{t+1}\rVert_1\big)$, and the hand-off is triggered when $0<D_t-D_{t+1}<\delta$ after a minimum number of steps. The method composes with caching (TeaCache) and FP8 attention (SageAttention), reporting up to $3.9\times$ speedups with small quality deltas; combined gains can reach $6.22\times$ on recent GPUs.

**Strengths:**

1. Originality. “Sketch-then-render” across noise regimes is a crisp insight.

2. Strong empirical speedups with minimal quality loss; synergy with TeaCache/FP8 attention.

3. Method, switch metric, and system composition are described concretely; diffusion equations are explicit.

**Weaknesses:**

1. Resource footprint. Requires hosting two models; memory and orchestration overheads on single-GPU edge devices need quantification.

2. Switch robustness. Generalization of the switch threshold $\delta$ across datasets/models is only partially explored.

3. Evaluation scope. Broader perceptual/temporal metrics and user studies would strengthen the evidence.

4. Reproducibility: missing details on schedulers and guidance settings that strongly affect quality/latency.

**Questions:**

1. What are peak VRAM and end‑to‑end latency (including encode/decode) under single‑ and multi‑GPU serving?

2. Can $\delta$ be auto‑calibrated from runtime signals (e.g., similarity proxies) instead of manual tuning?

3. How does performance change for long clips, high resolutions, and strong camera motion?

4. What happens if the small model’s capacity is substantially lower—does the handoff still preserve semantics?

---

> ### Author Response · Authors · 2025-11-20
>
> We would like to thank the Reviewer for the constructive feedback. Please find our responses below:
> 1. We report multi-GPU inference results in Appendix A.3. SRDiffusion integrates seamlessly with parallelization strategies such as Ulysses and Ring Sequence Parallelism, achieving additional speedup.
> 2. We believe this problem is inherently multi-objective. There is an unavoidable trade-off between generation quality and generation speed, which makes a threshold necessary. And exploring more flexible and learnable switching strategies would be an interesting direction for future work.
> 3. In Section 4.4, we present VBench scores, similarity metrics, and speedup results at a higher resolution (1280×720). These results demonstrate that SRDiffusion can be effectively applied to higher-resolution scenarios.

---

### Author Response · Authors · 2025-11-20
**General response and summary of changes in the revision**

We sincerely thank the reviewers for their constructive feedback. Based on your suggestions, we have made substantial improvements to the manuscript. We revised and expanded the related work, including distillation-based acceleration methods, and improved clarity and reduced redundancy throughout the paper. And we have added three new experiments to validate and strengthen our claims:

1. Support timestep distillation models: We extended SRDiffusion to support FastWan (FastVideo Team), using FastWan2.1-14B for the first denoising step and switching to FastWan2.1-1.3B for subsequent steps. This achieves a 2.20× speedup while preserving semantic and visual quality, demonstrating SRDiffusion’s compatibility with distillation-based models and other acceleration methods.
2. User study: We conducted a human survey (Section 4.2) to compare perceptual quality: the win rate of Wan 14B vs. SRDiffusion is approximately 50%, indicating that any degradation is nearly imperceptible; and SRDiffusion actually outperforms TeaCache in perceived quality.
3. Multi-GPU inference results: demonstrating that SRDiffusion integrates seamlessly with parallelization strategies such as Ulysses and Ring Sequence Parallelism, achieving further acceleration.

We refined the explanation of our core motivation: large models show significant advantages in semantic understanding, while their gains in visual quality are relatively limited. VBench’s semantic metrics reliably capture this trend, and SRDiffusion preserves the semantic performance of large models while introducing only minor, subjective visual variations. And, we have uploaded 20 prompt-based video generation samples to Google Drive for your reference. (https://drive.google.com/drive/folders/1zyKouSuulfkwtRT9pNx_u_MJLCLnY4D6?usp=sharing)

I believe the contributions worth highlighting to the community are as follows:
1. We propose an inference approach that combines large and small models, which preserves the semantic performance of large models while introducing only minor, subjective visual variations.
2. Our work also provides evidence that different denoising stages require different model capacities and ablity, suggesting that different weights should be used at different stages, e.g., for improving MoE architectures (the design of Wan2.2 was inspired by our findings).
3. We propose a new paradigm for video generation services: providers can offer a Sketching Model API, while the final video generation is completed on the client side using smaller models or different fine-tuned models.

---

### Author Response · Authors · 2025-12-03
**Summary for Area Chair**

Dear Area Chair,

We sincerely thank you and the reviewers for the thoughtful and constructive feedback. In response, we have substantially improved the manuscript in terms of clarity, completeness, and evaluation. Key revisions and response include:

1. Enhanced Related Work: Expanded discussion of distillation-based and other acceleration methods.
2. We add new experiments based on the reviewer feedback:
    * Support for timestep distillation models: Integrated SRDiffusion with FastWan (FastVideo Team), achieving 2.20× speedup while preserving semantic and visual quality by using the 14B model only for the first denoising step.
    * User study (Section 4.2): Human evaluators perceive no significant quality drop. Wan 14B vs. SRDiffusion shows ~50% win rate and SRDiffusion outperforms TeaCache in perceptual quality.
    * Multi-GPU inference (Appendix A.3): Demonstrated seamless compatibility with Ulysses and Ring Sequence Parallelism for further acceleration.
3. Clarified Core Motivation: Large models excel in semantic understanding but offer diminishing returns in visual fidelity. SRDiffusion preserves semantic strength while incurring only minor, subjective visual variations. We validated SRDiffusion via VBench, similarity metrics, and human evaluation.
4. Practical Deployment Insights:
    * Switching between models incurs negligible overhead (only one extra VAE decode/encode when needed).
    * A Sketching–Rendering Disaggregation design enables deployment on heterogeneous hardware (e.g., high-end GPUs for sketching, edge devices for rendering).
5. Broader Applicability:
    * Supports cross-family cooperation (e.g., Wan + CogVideo, Wan + Vchitect2.0) without retraining.
    * Enables a training-free, flexible “Sketching Model API” paradigm for video generation services.

We have also uploaded 20 prompt-based video samples to Google Drive for qualitative assessment: https://drive.google.com/drive/folders/1zyKouSuulfkwtRT9pNx_u_MJLCLnY4D6?usp=sharing

Importantly, SRDiffusion can acceleration 3x with nearly no model performance loss is orthogonal to existing acceleration techniques (e.g., caching, quantization, distillation) and provides evidence that different denoising stages benefit from different model capacities, a insight already influencing some model design (e.g., Wan2.2).

We believe these contributions, including such huge acceleration and insight, adaptive model switching, training-free cross-model collaboration, which can offer significant value to the community.

Thank you once again for your time and insightful feedback.

---

### Meta-Review · Area_Chair_aw8A · 2025-12-21

**Summary:**

This paper proposes “SRDiffusion”, a training-free framework to accelerate video diffusion inference. The core idea behind this paper is: first utilize a large, computationally model for the early high-noise steps and then switch to a smaller, faster model for the later low-noise steps. Experiments are conducted primarily on Wan and CogVideoX models using VBench, claiming speedups of 2-3 times with minimal quality degradation.

However, the reviewers raise several core concerns including: (1) Limited Novelty and Incremental Nature, (2) Practical Deployment and Memory Overhead., (3) Unconvincing Motivation and Reliance on VBench, and (4) Dependence on Model Availability.

In summary, I recommend the submission for Reject. I strongly encourage the authors to further polish the contents accordingly

**Reviewer Concerns:**

Nearly all four foregoing core concerns are still (partially) outstanding, while the authors' rebuttal addressed some minor concerns.


Addressed Concerns:

(1) More comprehensive empirical analysis based on the reviewer feedback:
- Support for timestep distillation models: Integrated SRDiffusion with FastWan (FastVideo Team), achieving 2.20× speedup while preserving semantic and visual quality by using the 14B model only for the first denoising step.
- User study (Section 4.2): Human evaluators perceive no significant quality drop. Wan 14B vs. SRDiffusion shows ~50% win rate and SRDiffusion outperforms TeaCache in perceptual quality.
- Multi-GPU inference (Appendix A.3): Demonstrated seamless compatibility with Ulysses and Ring Sequence Parallelism for further acceleration.
(2) Clarified Core Motivation: Large models excel in semantic understanding but offer diminishing returns in visual fidelity. SRDiffusion preserves semantic strength while incurring only minor, subjective visual variations. The authors validated SRDiffusion via VBench, similarity metrics, and human evaluation.
(3) Broader Applicability:
- Supports cross-family cooperation (e.g., Wan + CogVideo, Wan + Vchitect2.0) without retraining.
- Enables a training-free, flexible “Sketching Model API” paradigm for video generation services.

Outstanding concerns:

(1) Limited Novelty: Reviewer mRpW, j72M and gjjr all raised significant concerns regrading the novelty of the approach. The concept of using different models for different noise levels is not new (e.g., SDXL-refiner, Wan2.2). While the authors argue their goal is acceleration during inference rather than quality enhancement, the technical contribution remains a heuristic switching strategy between two pre-existing models.

(2) Dependence on Model Availability: As noted by Reviewer gjjr, the method is not truly universal, it strictly requires the existence of a small version of a specific model family. This limits the generalizability of the contribution compared to truly model-agnostic acceleration techniques.

(3) Unconvincing Motivation and Reliance on VBench: Reviewer gjjr reised a critical point regarding the paper's premise. The authors justify the method by showing that Wan 14B and 1.3B have similar VBench scores. The reviewer correctly notes that if the small model is truly that close to the large model in quality, the utility of the large model is questionable. Conversely, if VBench fails to capture the perceptual gap that clearly exists between 14B and 1.3B models, then the entire evaluation is flawed. The rebuttal`s inclusion of a small user study was not sufficient to overturn the skepticism that the method relies on a metric that may be insensitive to the specific visual degradations caused by switching models.

(4) Practical Deployment and Memory Overhead: Reviewer mRpW and 6FUF highlighted the resource implications of maintaining two models in memory. While the authors argue that an 80GB GPU can hold both Wan 14B and 1.3B, this introduces significant orchestration complexity and hardware requirements. For many practical acceleration scenarios, enlarging the VRAM requirement to achieve latency reduction is a poor trade-off compared to other acceleration methods.

**Reviewer Scores:**

Reviewer 6FUF: 6 -> 6
- Justification: The reviewer is the most positive. However, Reviewer 6FUF engages with the severe practicality concerns raised by gjjr regarding the memory footprint vs. speed trade-off. No responses during the rebuttal.

Reviewer mRpW: 4 -> 4
- Justification: The reviewer maintains his score, and no responses during the rebuttal. However, the reviewer notes that the idea is incremental and relies on imperfect proxies for perceptual quality.

Reviewer j72M: 4 -> 6
- Justification: This reviewer tended to raise the score after the rebuttal. The authors have partially addressed the reviewer`s questions. However, given  the reviewer's strong initial criticism regarding “Limited Novelty” and “Insufficient Related Work”, it is probable they would have only raised it to 4.

Reviewer gjjr: 2 -> 2
- Justification: This reviewer remained unconvinced by the rebuttal, citing “Limited Novelty”, “Unconvincing Motivation”, and “Redundant Writing”. I think the reviewer tends to maintain their score as 2.

---

### Decision · Program_Chairs · 2026-01-26

Reject